# Lipocalin 2 Deficiency Restrains Aging-Related Reshaping of Gut Microbiota Structure and Metabolism

**DOI:** 10.3390/biom11091286

**Published:** 2021-08-28

**Authors:** Xiaoxue Qiu, Chi Chen, Xiaoli Chen

**Affiliations:** Department of Food Science and Nutrition, University of Minnesota, Twin Cities, MN 55455, USA; qiuxx168@umn.edu (X.Q.); chichen@umn.edu (C.C.)

**Keywords:** lipocalin 2, gut microbiota, inflammation, aging

## Abstract

Gut microbiota modulate age-associated changes in metabolism, innate immune responses, and cognitive function. However, the involvement of host factors in the regulation of age-dependent gut microbial structure and intestinal inflammation is largely unknown. Lipocalin 2 (Lcn2) has previously been identified as an adipocytokine and characterized as an important regulator of diet-induced obesity and inflammation. Previous studies have shown that Lcn2 plays a role in high fat diet-induced reshaping of gut microbiota and intestinal inflammation. However, the role of Lcn2 in the regulation of aging-related reshaping of gut microbiota is unclear. Herein, we demonstrate that fecal levels of Lcn2 are reduced during aging. Age reshaped gut microbiota composition in wild-type (WT) mice. Interestingly, Lcn2 deficiency diminished this effect of aging in Lcn2 knockout (LKO) mice, leading to decreased bacterial diversity and increased Firmicutes to Bacteroidetes (F to B) ratio. Specifically, we identified 16 bacteria at the family level that were differentially abundant between WT and LKO mice at old age. Several health-promoting bacteria, including SCFA-producing bacteria, were significantly less prevalent in old LKO mice compared to WT mice, indicating that Lcn2 deficiency shifts the aging-related gut microbial community towards an unhealthy population and lowers microbial butyrate production. Our results provide a line of evidence that Lcn2 plays a role in the control of aging-related reshaping of gut microbiota composition and metabolites.

## 1. Introduction

For decades, the aging process has been characterized as a progressive attenuation of the host’s ability to sustain efficient immune responses as well as metabolic and neurological health. Until recently, efforts in cellular and molecular research, for example, the identification of cellular senescent pathways, enabled us to better understand various mechanisms that explain the complicated processes of age-associated inflammation, metabolic dysfunction, and neurological disorder [1]. In large human cohort studies, remarkable gut microbiota remodeling was observed in elderly populations over 65 years old [2,3,4,5,6,7]. However, the interpretation of the data has been challenged because of a wide range of interfering lifestyle factors including diet, medications, exercise, place of residence, and genetic background. Given that gut microbiota are closely related to various physiological features including gut barrier function, intestinal inflammation balance, metabolism, and gut–brain axis [8], it is believed that the age-related perturbations in gut microbiota composition and function could play an important contributing role in the increased frailty of elderly people. Age-related reduction in chew ability, digestion, and intestinal transit time may affect the dietary choice and food digestion, which could contribute to the alterations in gut microbiome in elderly subjects.

As an “organ”, gut microbiota can produce bacterial metabolites and microbial-associated molecular patterns (MAMPs) such as lipopolysaccharides to regulate host metabolism and inflammation [9,10,11]. Short-chain fatty acids (SCFAs) are a well-studied example of gut microbiota-derived metabolites. A line of evidence shows that the loss of gut microbial diversity occurs during aging and affects the aging process [12]. In a recent study, aging-specific changes in colonic gene expression were strongly correlated with specific bacteria genera [13]. However, how gut microbiota composition is reshaped during aging and which host factors are involved remain unknown. 

Lipocalin 2 (Lcn2) has been shown to play an important role in metabolic inflammation in obesity and metabolic diseases. Our recent study reported that adipose tissue-specific overexpression of Lcn2 protected against age-related metabolic deterioration [14]. Moreover, increasing evidence has indicated the involvement of Lcn2 in the regulation of reshaping gut microbiota in inflammatory bowel disease (IBD) and colitis-associated intestinal inflammation [15,16]. Indeed, Lcn2, as an antimicrobial peptide, is able to suppress the growth of siderophore-dependent bacteria. Our recent studies demonstrated that Lcn2 deficiency reshapes gut microbiota and remodels the microbiota-derived metabolome, including decreased production of short-chain fatty acids (SCFAs) and SCFA-producing microbes during diet-induced obesity [17]. However, the role of Lcn2 in age-related reshaping of gut microbiota remains unknown. 

In this study, we reported for the first time that aging regulates intestinal Lcn2 expression and secretion into the gut lumen in an age-dependent manner. While Lcn2 expression is induced in the colon, its secretion is disrupted, leading to lower fecal Lcn2 concentrations during aging. Lcn2 deficiency alters age-related gut microbial structure, specifically lowers the prevalence of health-beneficial bacteria, and reduces microbial butyrate production. Overall, this study highlights the protective role of Lcn2 against age-related gut microbial dysbiosis. 

## 2. Materials and Methods

### 2.1. Animals

C57BL/6J mice were purchased from Jackson Laboratory (Bar Harbor, ME, USA). Wild-type (WT) and global Lcn2 knockout (LKO) mice were maintained on a C57BL/6J background by heterozygous breeding scheme, as previously described (protocol ID: 1407-31711A) [18]. Animals were housed with a 12 h light–dark cycle in a specific pathogen-free facility at the University of Minnesota. Male mice were weaned at the age of 4 weeks and fed an RCD until the end of the experiment. Another set of male mice at the age of 45 weeks was fed an HFD (fat calories: 60% lard; Bio-Serv F3282; New Brunswick, NJ, USA) for 8 weeks. Age-matched male mice were allocated into cages (1–3 mice/cage). Animal handling followed the National Institutes of Health guidelines.

### 2.2. Relative Quantitative RT-PCR

Total RNA was extracted from gut tissue samples using TRIzol (ThermoFisher Scientific, Waltham, MA, USA). cDNA was synthesized using High Capacity cDNA RT kit (ThermoFisher Scientific, Waltham, MA, USA). Quantitative PCR was conducted using PowerUP™ SYBR™ Green Master Mix (ThermoFisher Scientific, Waltham, MA, USA). mRNA levels were determined by StepOne Real-Time PCR system (Applied Biosystem Foster City, CA, USA), using the ΔΔCt method to calculate the results. TATA box binding protein 2 (*TBP**2*) gene was used as the internal reference gene. Primer sequences are shown in Appendix A.

### 2.3. Western Blotting

Gut tissue samples were homogenized and lysed using radioimmunoprecipitation assay (RIPA) buffer (Sigma-Aldrich, Inc., St. Louis, MO, USA) supplemented with protease inhibitors. Protein concentrations were measured using the bicinchoninic acid (BCA) method. Equal amounts of proteins were loaded and separated on 10% SDS–PAGE, transferred to a nitrocellulose membrane and immunoblotted with indicated antibodies. Blots were visualized by enhanced chemiluminescence (ThermoFisher Scientific, Waltham, MA, USA). Antibodies used in this study were as follows: anti-Lcn2 (R&D System, Minneapolis, MN, USA), anti-NFκB (Santa Cruz Biotechnology, Inc., Dallas, TX, USA), anti-phospho-NFκB, and anti-β-actin (Cell Signaling Technology, Danvers, MA, USA). 

### 2.4. Quantification of Fecal Lcn2 by ELISA

Fecal samples were homogenized following a protocol from a previous study [16]. Briefly, frozen stool samples were reconstituted in 10 volumes (v (μL)/w (mg)) of PBS containing 0.1% Tween 20 and protease inhibitors, followed by vortexing for 15 minutes to obtain a homogenous fecal suspension. After 10 min centrifugation at 12,000 rpm under 4 °C, clear supernatants were collected and stored at −80 °C until analysis. Lcn2 levels in the supernatants were determined using mouse NGAL ELISA kit (BioLegend, San Diego, CA, USA). 

### 2.5. Isolation of Bacterial DNA and 16S rRNA Sequencing

Stool samples were collected from mice, snap-frozen in liquid nitrogen, and stored at −80 °C until use. Total fecal DNA was extracted using a DNA stool mini kit (QIAGEN). Bacterial 16S rRNA were amplified using primers (515F and 806R) for the V4 region. After quantification, the PCR products were sequenced using paired-end 300 bp with Ilumina MiSeq platform at the University of Minnesota Genomics Center. 

### 2.6. 16S rRNA Microbiome Analysis

Read quality was assessed using DADA2 software in R and then analyzed using the DADA2 pipeline. Samples were filtered and trimmed using filterAndTrim() with the following settings: truncLen = c(250, 200), maxN = 0, maxEE = c(3,3), truncQ = 2, and rm.phix = TRUE. Reads were denoised and dereplicated, and sample composition was inferred using the default DADA2 settings. Chimeric amplicon sequence variants (ASVs) were removed with removeBimeraDenovo() with method = “consensus”. The Silva database version 132 was used for taxonomic and species assignment. The ASV sequences were extracted and aligned to form a phylogenetic tree for downstream diversity analyses [19]. Phyloseq was used for alpha and beta diversity metric calculations and plot generation [20]. Beta diversity analyses—CoA using unweighted UniFrac distance metrics were ordinated with normalized ASVs [21]. ASV normalization was performed by dividing raw ASV counts by total ASV counts to give proportional abundances [22]. 

### 2.7. Determining Differentially Abundant Microbes

Differentially abundant microbes were determined as previously described [17]. Briefly, raw ASV counts were extracted from the phyloseq object for differential microbe abundance analysis in MASS package in the R environment (version 4.0.0). Normalized ASVs that had low variance (variance < 1 × 10^−6^) were removed from the raw ASV counts matrix. These criteria resulted in 241 ASVs with taxa names remaining for downstream analysis using a negative binomial generalized linear model (glm-nb, *read count* ~ -1 + geno:diet + offset, for each of 241 ASVs) with the log-link function. Comparison of microbe abundance between genotypes was performed as Z-tests with mean values and standard errors estimated from the fitted model. The two-tailed *p* values obtained from all the comparisons were adjusted to *q* values using the qvalue package via a false discovery rate (FDA)-based multiple testing correction procedure [23]. ASVs were considered differentially abundant if they had log2 (fold change) ≥ 1 fold and *q* ≤ 0.05 in LKO compared with WT genotype. Heatmaps of differentially abundant microbes were plotted using pheatmap package (version 1.0.12) with row scaling. 

### 2.8. Histology and Immunohistochemistry 

Gut tissues were fixed in 10% neutral buffered formalin with the 1:10 ratio of tissue to formalin overnight. After complete fixation, tissues were transferred into 70% ethanol. Paraffin embedding and hematoxylin and eosin (H&E) staining were performed by the Comparative Pathology Shared Resource at the University of Minnesota. Immunohistochemistry (IHC) was carried out using VECTASTAIN Elite ABC-HRP Kit (ThermoFisher Scientific, Waltham, MA, USA) and ImmPACT DAB Peroxidase Substrate Kit (ThermoFisher Scientific, Waltham, MA, USA) following the manufacturer’s instructions. Antigen retrieval was conducted in citric buffer using IHC-Tek Epitope Retrieval System (ThermoFisher Scientific, Waltham, MA, USA). Anti-CD11c (97585S) from Cell Signaling Technology (Cell Signaling Technology, Danvers, MA, USA) was used for IHC. Olympus BX53 microscope was used for image acquisition. 

### 2.9. Metabolomics Analysis of Fecal Contents

Fecal samples were soaked in 10 volumes (v (μL)/w (mg)) of 50% acetonitrile (ACN) overnight at 4 °C and then homogenized by vortex and sonication for 10 min. After centrifugation at 18,000× *g* for 10 min, the supernatants were obtained for derivatization reactions. For detecting fatty acids, the samples were derivatized with 2-hydrazinoquinoline (HQ) prior to the LC-MS analysis [24]. Briefly, 2 μL of fecal supernatants was added into 100 μL freshly prepared ACN solution with 14.6 μM d4-acetic acid (internal standard for SCFA) and 0.286 μg/mL ^13^C2-palmitic acid (internal standard for LCFA) containing 10 mM TPP, 10 mM DPDS, and 10 mM HQ. The reaction system was incubated at 60 °C for 30 min, followed by quickly chilling on ice, mixing with 100 μL ice-cold deionized water, and centrifuging at 18,000× *g* for 10 min. The supernatants derived from HQ reaction were transferred into HPCL vials for LC–MS analysis using an Acquity ultraperformance liquid chromatography (UPLC) system (Waters, Milford, MA, USA). Samples were separated in a BEH C18 column with gradient mobile phases ranging from water to 95% aqueous ACN containing 0.1% formic acid in a 10 min run. MassLynx™ and SIMAC-P+™ software (Waters, Milford, MA, USA) were used for mass chromatograms and mass spectral data acquisition and procession. 

### 2.10. Statistical Analysis

All results are shown as mean ± SEM. All data were assessed for homogeneity of variance by Levene’s test and for normality by Shapiro–Wilk test in the R environment (version 4.0.0). One-way ANOVA, two-way ANOVA followed by post hoc Tukey’s, or two-tailed unpaired Student’s t-test was used to determine the differences between groups. An adjusted *p*-value of less than 0.05 was considered statistically significant. Heatmaps were plotted using the pheatmap package (version 1.0.12) with row scaling. All above statistics tests were performed using R (version 4.0.0). 

## 3. Results

### 3.1. Aging Reduces Lcn2 Secretion into the Gut Lumen

Previous studies focused mainly on the mucosal Lcn2 expression during colitis-associated intestinal inflammation [15,25]. Our published studies have demonstrated that HFD feeding regulates intestinal Lcn2 expression and secretion in a time-dependent manner [17]. A recent study reported that gut microbiota and colonic gene expression are profoundly changed during aging, which is associated with diminished colonic health [13]. These transcriptional perturbations could drive the development of age-related diseases, such as obesity, metabolic disorders, and neurodegenerative diseases. So far, little is known about how the aging process alters Lcn2 expression in the gut. To explore this, we examined Lcn2 expression in the colon and ileum of RCD-fed normal C57BL/6J male mice with the ages of 6 months (young), 12 months (middle-aged), and 18 months (old). Our time-course study showed that Lcn2 protein levels were not altered in the colons of middle-aged mice, but significantly increased (*p* = 0.0051) in the colons of old mice compared to young mice (Figure 1A,B). However, Lcn2 was not induced in the ileum as aging (Figure 1C,D). Interestingly, fecal Lcn2 levels were significantly decreased in aged mice compared with young mice, suggesting that Lcn2 secretion into the gut lumen decreases during aging (Figure 1E). 

Next, we determined whether the colonic Lcn2 expression in response to HFD-induced metabolic stress is modulated with the aging process. To that end, normal C57BL/6J male mice at the ages of 8 weeks (young) and 45 weeks (middle-aged) were fed an HFD for 8 weeks, and Lcn2 protein levels were determined in the colon. We found that HFD feeding was able to strongly induce Lcn2 expression in the colons of both young and middle-aged mice, which is consistent with our previous findings [17]. Interestingly, in response to HFD challenge, middle-aged mice expressed significantly lower levels of Lcn2 in the colon than young mice (Figure 1F,G), indicating that aging diminishes HFD-induced Lcn2 expression in the colon. Since Lcn2 serves as an inflammatory marker and has anti-inflammatory properties [16,25], age-related decline in HFD-induced Lcn2 expression suggests that the protective role of Lcn2 against metabolic stress in the context of intestinal homeostasis is disrupted with aging. 

### 3.2. Lcn2 Deficiency Reduces Gut microbial Diversity in Aged Mice

Lcn2, as an antimicrobial peptide, has been shown to preserve healthy gut microbial structure in a colitis mouse model [16,25]. To examine the role of Lcn2 in regulating gut microbiota composition during aging, we carried out a time-course study of aging on young, middle-aged, and old wild-type (WT) and Lcn2 knockout (LKO) mice. Fecal samples were collected at each stage of age. The richness of microbial diversity shown as Chao 1 was significantly decreased in aged LKO mice compared with that in aged WT mice (Figure 2A). Shannon, another microbial diversity metric that represents both the richness and evenness of bacteria, was also significantly decreased in LKO mice compared to WT mice at old age. (Figure 2A). All the above results suggest that LKO mice harbor fewer bacterial species and have imbalanced microbial distribution compared to WT mice when they age. In WT mice, the number of bacterial species was significantly increased, whereas this increase was not observed in LKO mice. On the contrary, LKO mice experienced reduced microbial diversity from middle to old age (Figure 2A). Principal coordinate analysis (Weighted Unifrac) showed that aging shifts gut microbial community in WT, but not in LKO, mice (Figure 2B,C), suggesting that Lcn2 is involved in aging-induced reshaping of overall gut microbial structure. As expected, WT and LKO mice displayed similar microbial structure at both young and middle-aged stages. However, dissimilarity in microbiota composition between WT and LKO mice appeared at old age (Figure 2D). Taken together, our findings reveal an important role of Lcn2 in age-related reshaping of gut microbial structure.

### 3.3. Lcn2 Regulates Specific Bacterial Abundance in an Age-Dependent Manner

Profound shifts in bacterial taxa have been reported in the fecal microbiome of elderly populations compared to children and young adults, and these shifts have been associated with frailty and health status in seniors [2,6,7,26]. It is believed that host genetics are an interfering factor contributing to the age-related shaping of gut microbial taxonomic diversity. To further support this idea, we performed taxonomic diversity analysis on WT and LKO mice in a time-course of aging study. We found that the ratio of Firmicutes to Bacteroidetes (F to B ratio) was much higher in LKO mice compared to WT mice, which occurred only in old age (Figure 3A). Interestingly, the F to B ratio was decreased in old WT mice compared to young and middle-aged WT mice, whereas this age-dependent change was not observed in LKO mice (Figure 3A).

According to the generalized linear model of genotypic effect [27], we identified bacteria with abundances that were specifically altered by Lcn2 deficiency. At the family level, the abundances of four bacteria (Clostridiales, Deferribacteraceae, Helicobacteraceae, and Marinifilaceae) were significantly increased in LKO mice (Figure 3B). Strikingly, this increase occurred only at young and middle age, and there was no increase in old LKO mice compared to WT controls (Figure 3B). We also found that 12 bacteria were significantly decreased in LKO mice in an age-dependent manner (Figure 3B). Interestingly, 10 of them, including four major families (Ruminococcaceae, Muribaculaceae, Rikenellaceae, and Helicobacteraceae) and six minors (Prevotellaceae, Bifidobacteriaceae, Saccharimonadaceae, Marinifilaceae, Deferribacteraceae, and Tannerellaceae) were decreased only at old age, and the other two bacteria, Atopobiaceae and Desulfovibrionaceae were significantly reduced in LKO mice at young and middle age, respectively (Figure 3B). Of those ten bacteria that were decreased in aged LKO mice, several are health beneficial. For example, Ruminococcaceae and Bifidobacteriaceae are health-promoting bacteria, and the latter functions as a probiotic formulation [28,29]. Muribaculaceae has been reported to be an SCFA producer [30]. More interestingly, three bacteria (Deferribacteraceae, Clostridiales, and Marinifilaceae) thrived in LKO mice prior to aging, but their growth was suppressed in aged LKO mice (Figure 3B). These findings suggest that Lcn2 deficiency worsens the perturbations in the abundances of specific gut microbes of mice at old age. 

Lastly, we used a linear model to perform the analysis of amplicon sequence variants (ASVs) that represent bacterial species with unknown names. Results showed that WT and LKO mice displayed a completely different ASV distribution pattern regardless of age (Appendix A). Additionally, we identified 67 increased and 132 decreased ASVs in LKO mice compared to WT mice (Appendix A). Taken together, these data reveal that Lcn2 plays a role in the control of gut microbiota homeostasis at the stage of old age. 

### 3.4. Lcn2 Deficiency Has No Impact on Intestinal Inflammation during Aging

The crosstalk between gut microbiota and intestinal inflammation has been well documented. We showed that Lcn2 has a profound role in regulating gut microbiota composition in old mice (Figure 2 and Figure 3). To investigate the possible mechanisms by which Lcn2 modulates microbial structure during aging, we examined the state of intestinal inflammation in WT and LKO mice at the age of 18 months. The mRNA expression levels of inflammatory cytokines in the colon, including proinflammatory (TNFa, IL-17, and IFNγ) and anti-inflammatory (IL-10) cytokines, were significantly upregulated in old compared to young (3-month-old) WT mice (Figure 4A), suggesting that intestinal inflammation increased with aging. However, the mRNA expression levels of these pro- and anti-inflammatory cytokines (Figure 4A) as well as phosphorylated NFκB levels (Figure 4B,C) were not different between WT mice and LKO mice at old age. Moreover, histological analyses indicated that WT and LKO mice had similar overall intestinal architecture at both young and old age (Figure 4D). While more CD11c+ cells were seen in the colon in LKO mice compared to WT mice at young age (Figure 4E,F), the number of CD11c+ cells was not different between WT and LKO mice at old age. These data suggest that Lcn2 deficiency does not seem to affect aging-induced inflammation in the colon, and that alterations in microbiota composition in old LKO mice are independent of Lcn2 deficiency’s effect on intestinal inflammation.

### 3.5. Lcn2 Deficiency Reduces Microbial Butyrate Production in Old Mice

To better understand how Lcn2 regulates the function of gut microbiota during aging, we assessed and compared the composition of fecal metabolites between WT and LKO mice at young, middle, and old ages. The projections to latent structures-discriminant analysis (PLS-DA) model showed that WT and LKO mice had a similar structure of fecal metabolites at the young age (Figure 5A). The structures of fecal metabolites from middle-aged and old mice were clearly separated from those from young mice in both genotypes (Figure 5A). No further separation of fecal metabolite structure was observed between middle and old age in WT mice (Figure 5A). Interestingly, fecal metabolites clustered differently between middle and old in LKO mice, leading to a clear shift of fecal metabolites between WT and LKO mice at middle age (Figure 5A). These results indicate that the effect of Lcn2 deficiency on gut microbial metabolism is age dependent. Fecal concentrations of SCFAs (acetic and propionic acids) and long-chain fatty acids were comparable between WT and LKO mice at all stages of age (Figure 5B,C and Appendix A). However, fecal levels of linoleic acid (C18:2), which is polyunsaturated omega-6 fatty acid associated with increased inflammation and blood pressure, were significantly higher in aged LKO than aged WT mice (Appendix A). Intriguingly, we found that fecal butyric acid levels were significantly decreased in LKO mice compared to those in WT mice at old age (Figure 5D). During aging, fecal butyric acid levels were increased in WT mice, but this increase was absent in LKO mice at old age (Figure 5D). SCFAs not only serve as energy sources, but also signals regulating energy metabolism such as energy intake [31,32]. FFAR2 and FFAR3, as G-protein coupled receptors, functions as sensors for luminal SCFAs [33], and their expression is largely regulated by metabolic stress [34,35]. However, we did not observe changes in the mRNA expression levels of *FFAR2* and *FFAR3* in the colon during aging (Figure 5E)*,* suggesting that aging is not a regulator of SCFA receptors expression in the gut. Interestingly, Lcn2 deficiency caused a reduction in mRNA expression of these two receptors in the colon of old mice but not young mice, suggesting that Lcn2 regulates the SCFA signal pathway.

## 4. Discussion

Age-associated alterations in colonic gene expression are associated with increased intestinal permeability, modulated gut microbiota composition, and increased systemic inflammation [13,36]. In this study, gut inflammation was increased, as evidenced by upregulated expression of cytokines and increased numbers of CD11+ cells, in WT mice during aging. Similar to our observations in HFD feeding studies [17], the expression of Lcn2 in the colon was increased, but Lcn2 secretion into the gut lumen was decreased, in old mice compared to young mice, suggesting that the secretory pathways of Lcn2 into the gut lumen are impaired with aging. More interestingly, aging significantly attenuated HFD induction of Lcn2 expression in the gut. Since Lcn2 has an anti-inflammatory role and increased Lcn2 is beneficial for metabolic health [18,37], reduced induction of Lcn2 may explain why elderly individuals display defective immune responses when challenged with HFDs or pathogen infections, which renders them vulnerable to metabolic and infectious diseases. 

As lumen Lcn2 plays a critical role in maintaining gut microbiota symbiosis [17], age-related reduction in fecal Lcn2 may contribute to the disruption of microbiota homeostasis during aging. Similarly to what has been observed in HFD-induced obesity, aging can induce the development of microbial dysbiosis. However, the specific features of age-associated microbial dysbiosis have not been previously reported. We observed that WT old mice had increased bacterial diversity and decreased F to B ratio when compared to WT young mice. In human studies on age-related changes in gut microbiome, the F to B ratio was reported to be increased from infants to adults, followed by a decrease in elderly individuals [38]. These results support our findings that the F to B ratio declined in WT mice with aging. Interestingly, this dynamic change in F to B ratio disappears, and the F to B ratio remains higher, in LKO mice. 

Many studies have suggested that higher microbiota diversity is correlated with better health in adults, and that the loss of microbial diversity is associated not with chronological aging, but with increased frailty and reduced cognitive performance [12]. In the present study, microbial diversity increased with aging in WT mice, but this age-related increase diminished in LKO mice. Instead, microbial diversity was significantly decreased in LKO mice compared to WT mice at old age. This suggests that Lcn2 is important for maintaining higher levels of microbial diversity in mice at old age. It is unknown why microbial diversity increases with aging. Phylogenetic composition might be more useful for data interpretation when considering the effect of aging on gut microbiota. Regardless, we did observe that Lcn2 deficiency led to lower microbial diversity and higher F/B ratio in old mice. Our data strongly suggest that Lcn2 is involved in the regulation of temporal dynamics of gut microbiota during aging.

In an attempt to identify specific Lcn2-regulated bacteria during aging, we found that Lcn2 deficiency suppressed the growth of 12 family bacteria, particularly at old age. Specifically, health-promoting bacteria such as probiotic bacteria Bifidobacteriaceae [29], plant-fiber degradation bacteria Ruminococcaceae [28,39] and Prevotellaceae [40], and SCFA-producing bacteria Muribaculaceae [30], are potential Lcn2-regualted bacteria, as their abundances were significantly lower in LKO mice compared to WT mice at old age. The potential regulatory mechanism of Lcn2 in gut microbiota has been suggested in previous studies to inhibit the expansion of siderophore-dependent bacteria [41]. Since no family of bacteria thrives in old LKO mice, it is possible that Lcn2 deficiency suppresses the aforementioned families of bacteria’s growth indirectly by promoting the growth of other bacteria at lower taxonomic levels through bacteria–bacteria interaction. 

The interplay between gut microbiota and the host intestinal immune system is necessary for gut homeostasis. However, the complex mutual effects make it difficult to determine which effect is the initial driving force. In our previous studies, we observed that Lcn2 deficiency accelerated the development of HFD-induced inflammation in the colon [17]. Unlike the HFD-fed condition, Lcn2 deficiency-caused changes in gut microbial structure were not accompanied by significant alterations in the magnitude of intestinal inflammation in aged mice, suggesting that intestinal inflammation is not the outcome of dysbiosis or vice versa in LKO mice. These data also support our findings that Lcn2 regulated microbiota homeostasis and intestinal inflammation via an independent mechanism in aged mice.

Microbiota-derived metabolites, as the functional readout of gut microbiota, serve as the linkage nodes between gut and other tissues to regulate energy metabolism, immune responses, and tissue-specific physiological functions. We showed that fecal butyric acid levels were significantly decreased in LKO mice compared to WT mice at 18 months of age. Intriguingly, we also found that the abundances of fiber-fermented bacteria were decreased in old LKO mice, suggesting a potential link between decreased fiber-fermented bacterial abundance and lower butyrate levels in LKO mice. Moreover, decreased mRNA expression levels of SCFA receptors *FFAR2* and *FFAR3* in old LKO mice further support the idea that there is a lower SCFA level and signaling activation in LKO mice. In addition to serving as an energy substrate, butyrate functions as a signal molecule in reducing energy intake and obesity via activating FFAR2/3 and promoting the secretion of glocagen-like peptide-1 (GLP-1) [33]. However, we did not see any significant difference in body weight between WT and LKO mice at young, middle, or old age (Appendix A), suggesting that the reduced fecal butyric acid levels had no effect on age-related body weight gain in LKO mice. 

In summary, our study suggests that Lcn2 deficiency promotes gut microbiota dysbiosis during aging, and that Lcn2 is able to directly regulate microbial structure in an age-dependent manner. This effect is likely independent of intestinal inflammation. Loss of Lcn2 suppresses the growth of health-beneficial bacteria at old age, leading to decreased production of butyrate in the gut. Taken together, our results indicate that Lcn2 plays a role in age-related reshaping of gut microbiota and its regulated metabolites. However, there are some limitations of this study. Since Lcn2 exerts its antibacterial function through binding to specific subclasses of bacterial siderophores and preventing bacterial uptake of iron for growth, it is likely that Lcn2 may only affect the specific group of bacteria that relies on iron-bound siderophores for iron acquisition. The iron status may also affect the effect of Lcn2 deficiency, which was not tested in this study. In future studies, it would be of interest to conduct experiments testing whether Lcn2 affects only siderophore-dependent bacteria and how dietary iron availability or systemic iron deficiencies influence the effect of Lcn2 deficiency on gut microbiota.

## Figures and Tables

**Figure 1 biomolecules-11-01286-f001:**
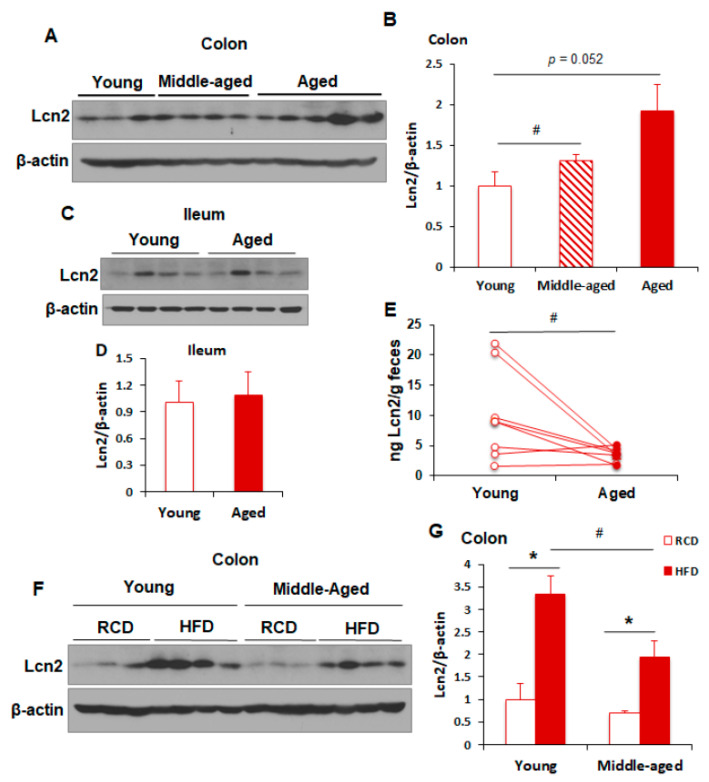
Regulation of Lcn2 expression and secretion in the gut during aging. (**A**,**B**) Male mice were fed an RCD and euthanized at the ages of 4 months (young), 10 months (middle-aged), and 18 months (old). Colon tissues were collected and homogenized in RIPA buffer. Lcn2 protein levels were determined by Western blotting (**A**) and quantified by ImageJ (**B**). (**C**,**D**) Male mice were fed an RCD and euthanized at the ages of 6 months (young) and 18 months (old). Ileum tissues were collected and homogenized in RIPA buffer. Lcn2 protein levels were determined by Western blotting (**C**) and quantified by ImageJ (**D**). (**E**) Fecal Lcn2 levels were measured using an ELISA kit. (**F**,**G**) Male mice at the ages of 2 months (young) and 11 months (middle-aged) were fed an HFD for 8 weeks. Colon tissues were collected for the detection of Lcn2 protein levels by Western blotting (**F**) and quantified by ImageJ (**G**). Data are presented as mean ± SEM. * *p* < 0.05 versus RCD-fed mice or as indicated. **^#^**
*p* < 0.05 versus young mice.

**Figure 2 biomolecules-11-01286-f002:**
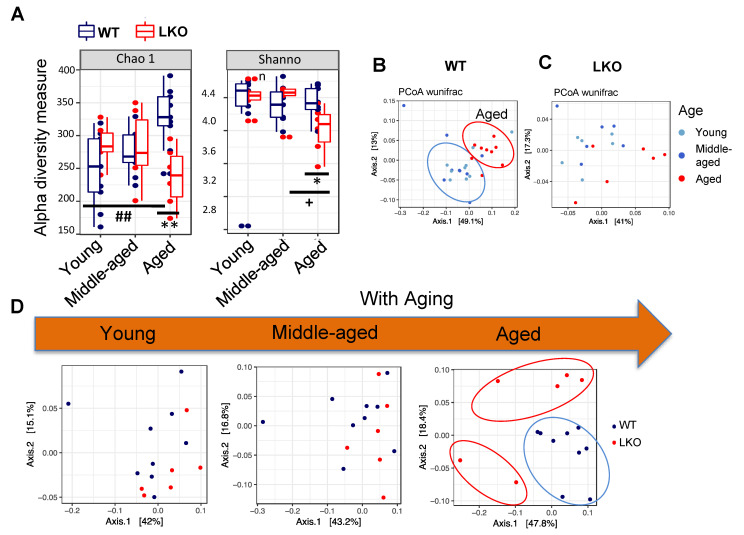
Age-associated reshaping of gut microbiota in LKO mice. Male WT mice (n = 9, 4 cages) and LKO mice (n = 6, 4 cages) were fed an RCD starting from weaning. Fecal samples were collected at the ages of 6 months (young), 12 months (middle-aged), and 18 months (old). (**A**) Alpha diversity metrics: Chao 1 and Shannon are shown to demonstrate the richness and evenness of gut microbiota communities. * *p* < 0.05, ** *p* < 0.01 versus WT mice. ^##^
*p* < 0.01 versus young WT mice. **^+^**
*p* < 0.05 versus middle-aged LKO mice. (**B**–**D**) Beta diversity (weighted unifrac) indicates the dissimilarity between WT (**B**) and LKO (**C**) mice at all three stages of age and the genotypic difference at each age stage (**D**).

**Figure 3 biomolecules-11-01286-f003:**
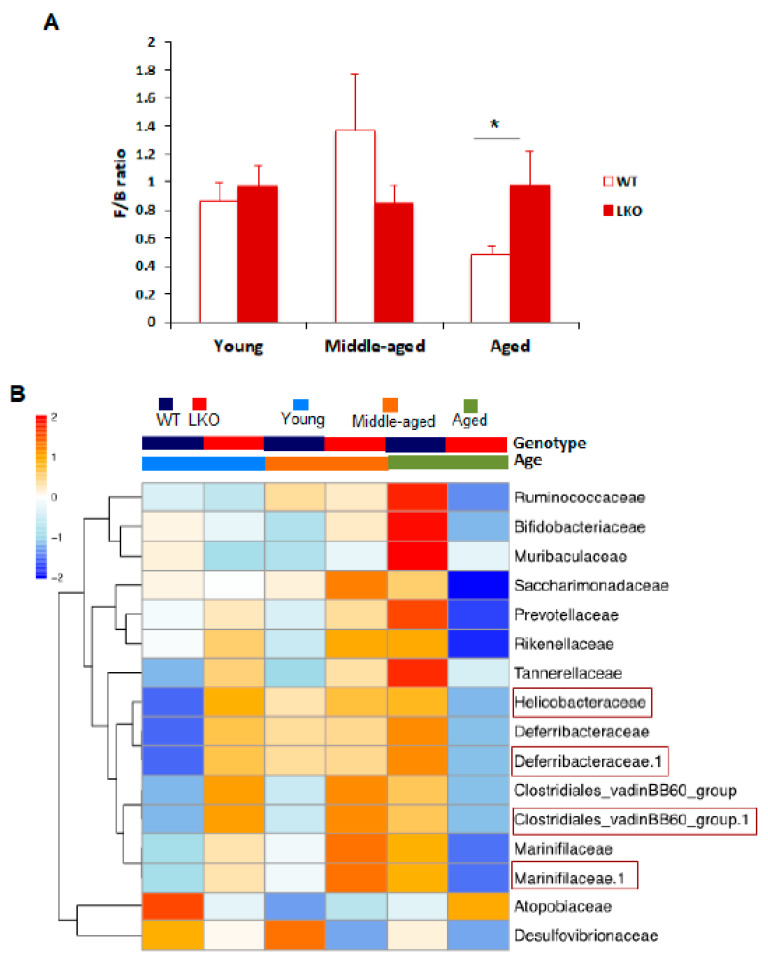
Identification of differentially abundant bacteria between WT and LKO mice during aging. (**A**) The ratio of Firmicutes to Bacteroidetes (F/B) in WT and LKO mice at all three stages of age. Data are presented as mean ± SEM (n = 6–9). * *p* < 0.05 versus old WT mice. * *p* < 0.01 versus young WT mice (**B**) Hierarchal clustering analysis (HCA)-based heatmaps on differentially abundant, including both increased and decreased, microbes at the family level between WT (n = 9, 4 cages) and LKO (n = 6, 4 cages) mice based on a negative binomial generalized linear model in the R environment. Highlighted bacteria were significantly more prevalent in LKO mice. *q* value ≤ 0.05 and log2 scale fold change ≥ 2 were the criteria to determine significance.

**Figure 4 biomolecules-11-01286-f004:**
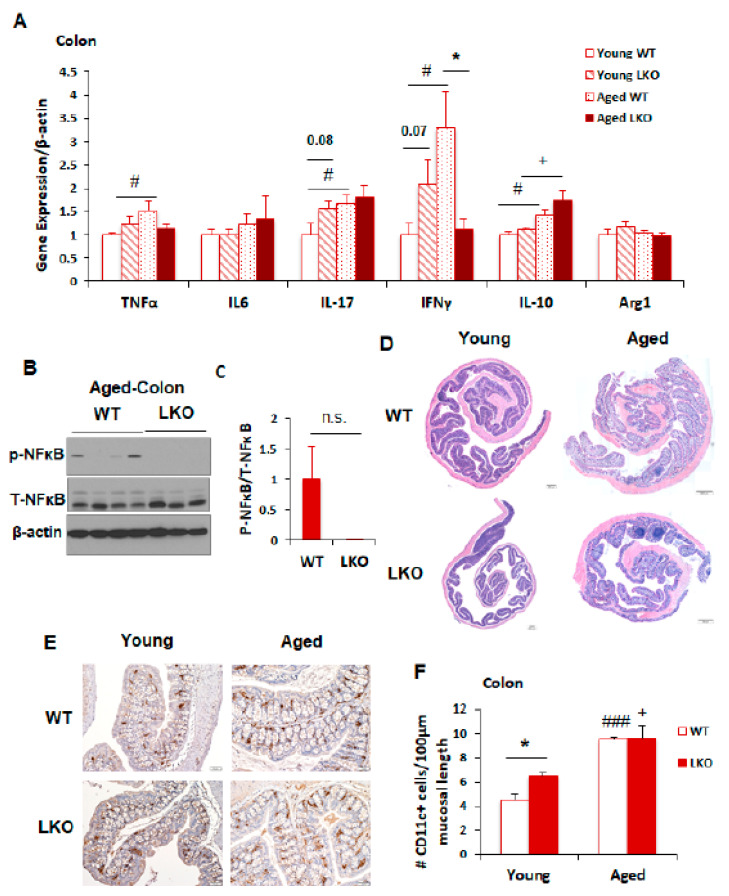
Lcn2 deficiency has no effect on intestinal inflammation during aging. RCD-fed male mice were euthanized at the ages of 6 months (young) and 18 months (old). (**A**) qPCR was performed to compare the mRNA expression levels of pro- and anti-inflammatory cytokines in the colon between WT and LKO mice. The results are presented as mean ± SEM (n = 5–9). (**B**,**C**) Colon tissues were homogenized and lysed in RIPA buffer for the examination of phospho-(S536)-NFκB and total NFκB levels between old WT and old LKO mice by Western blotting (**B**) and quantified by ImageJ (**C**). (**D**) H&E staining of colons from young and old WT and LKO mice. Scale bars = 200 μm, ×200. (**E**,**F**) Immunohistochemistry of CD11c on paraffin-embedded colon sections from young and old WT and LKO male mice. (**E**) Representative images of CD11c-stained colon sections, scale bars = 50 μm, ×200. (**F**) Results were quantified by counting the number of CD11c^+^ cells and measuring the mucosal length. Data are presented as mean ± SEM (n = 3) of the number of CD11c^+^ cells per 100 μm of mucosal length. * *p* < 0.05 versus WT mice. **^#^**
*p* < 0.05, **^###^**
*p* < 0.001 versus young WT mice. **^+^**
*p* < 0.05 versus young LKO mice.

**Figure 5 biomolecules-11-01286-f005:**
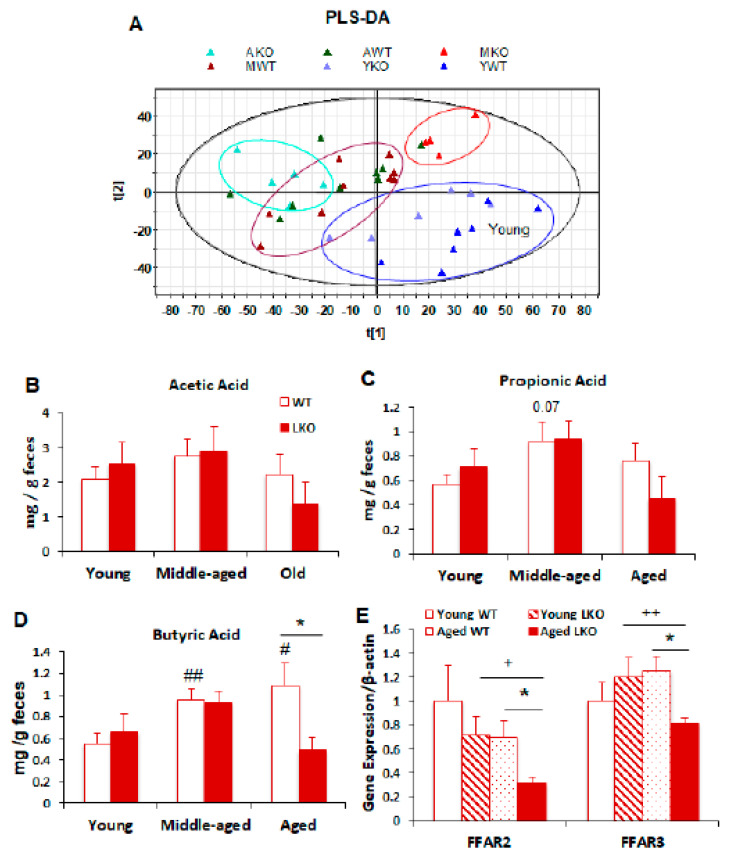
Lcn2 deficiency reduces fecal butyrate levels in old mice. Stool samples from WT and LKO at the ages of 6 months (young), 12 months (middle-aged), and 18 months (old) were homogenized in 50% ACN. (**A**) Data from LC–MS analysis of fecal extracts were processed by latent structures-discriminant analysis (PLS-DA) modeling. Shown is the scores plot of a PSL-DA model on the metabolites in fecal extracts. (**B**–**D**) Supernatants were collected for HQ reaction to detect SCFAs: acetic acid (**B**), propionic acid (**C**), and butyric acid (**D**). (**E**) mRNA expression levels of free fatty acid receptors *FFAR2* and *FFAR3* between WT and LKO mice at young and old age. Data are presented as mean ± SEM (n = 5–9). * *p* < 0.05 versus WT mice. **^#^**
*p* < 0.05, **^##^**
*p* < 0.01 versus young WT mice. **^+^**
*p* < 0.05, **^++^**
*p* < 0.01 versus young LKO mice.

## Data Availability

Data is contained within the article.

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
