# Peer review of "Lipocalin 2 Deficiency Restrains Aging-Related Reshaping of Gut Microbiota Structure and Metabolism"

_biomolecules, 2021, doi:10.3390/biom11091286_

Round 1

Reviewer 1 Report

In general, Qiu et al. is a well-written and clear presentation of fairly straightforward experimental results linking Lipocalin 2 (Lcn2/NGAL/Siderocalin/24p3) expression in an age-dependent manner to altered colonizing microbiota compositions. The results logically extend from prior work and advance the field.

However, two aspects of the work somewhat limit its impact.

First, the results are sometimes modest, achieving statistical significance but not with obviously dramatic effects. This is not a major concern, as this is sometimes how science works, particularly when studying complex phenotypes, but there is the concern that the authors are nibbling around the edges of what’s really going on.

However, the second concerning aspect is more disappointing. The hypothesis framing the work and the interpretation of the presented results are very limited, lacking real mechanistic insights, yielding a very descriptive report. The authors state “Lcn2, as an anti-microbial peptide is able to suppress the growth of siderophore-dependent bacteria”. This is, perhaps, an oversimplification of the extant literature and misses important aspects of Lcn2 function. The protein Lcn2 does not, on its own, have any anti-bacterial activity. Lcn2 binds to specific subclasses of bacterial siderophores, depriving affected bacteria of needed iron for growth. Lcn2 does not bind iron in the absence of siderophores, only binds certain siderophores, and only affects bacteria that rely on those specific siderophores for iron acquisition under iron-limiting conditions. It is not, therefore, surprising that Lcn2 expression levels in the gut could alter the composition of the colonizing microbiota, dependent on dietary iron availability or systemic iron deficiencies. The missed opportunities are (1) the lack of consideration of physiological iron levels, food/gut iron levels, and correlations with observed effects, and (2) correlating siderophore utilization of the affected bacterial species with the known ligand specificity of Lcn2.

In the absence of deeper mechanistic insights, the overall impact of the manuscript on the field remains real, but limited.

Reviewer 2 Report

Gut microbiota has been considered to be a key modulator for ageing-related metabolic and other disorders. However, how the factors secreted by other metabolically active tissues in the hosts affect the functions of the microbiota in gut needs to be further elucidated. In that context, Qiu et. al  reveal that an critical regulator during obesity named Lipocalin (Lcn2) tightly regulates ageing-related reshaping of gut microbiota in the manuscript. Particularly, the authors found that the levels of Lcn2 in feces are decreased during ageing, and ageing itself may change gut microbiota composition in the mice.  They further demonstrated that loss-of-function of Lcn2 causes decreased bacterial diversity and hence increased F to B ratio. Importantly, they identified some bacteria, including several health-promoting bacteria that were differentially varied in WT and Lcn KO mice during ageing, suggesting the key role of Lcn2 in shifting the aging-related gut microbial community.  The study is interesting and timely. The experimental designs are rigid, and the data achieved from the experiments are solid and informative. The manuscript is well written.  I do not have  major concerns. Minor suggestions are listed below for the authors:

1). Please define “F to B ratio” in the abstract;

2). “in chapter 2” in the main text is confusing. Please clarify the meaning;

3). Fig. 1A&F show Lcn2 increased under RCD for the middle-aged mice, while Fig. 1F shows no changes under the same conditions. Please clarify the results and keep them consistent.

4). Please discuss whether HFD might affect the results in Fig. 4 (No experiments are needed. Just discuss the possibility).
